# Ultrahigh-responsivity waveguide-coupled optical power monitor for Si photonic circuits operating at near-infrared wavelengths

Takaya Ochiai[1], Tomohiro Akazawa[1], Yuto Miyatake [1], Kei Sumita[1], Shuhei Ohno [1], Stéphane Monfray[2], Frederic Boeuf [2], Kasidit Toprasertpong [1], Shinichi Takagi[1] & Mitsuru Takenaka [1] ✉

A phototransistor is a promising candidate as an optical power monitor in Si photonic circuits since the internal gain of photocurrent enables high responsivity. However, state-of-the-art waveguide-coupled phototransistors suffer from a responsivity of lower than $10^3$ A/W, which is insufficient for detecting very low power light. Here, we present a waveguide-coupled phototransistor operating at a 1.3 μm wavelength, which consists of an InGaAs ultrathin channel on a Si waveguide working as a gate electrode to increase the responsivity. The Si waveguide gate underneath the InGaAs ultrathin channel enables the effective control of transistor current without optical absorption by the gate metal. As a result, our phototransistor achieved the highest responsivity of approximately $10^6$ A/W among the waveguide-coupled phototransistors, allowing us to detect light of 621 fW propagating in the Si waveguide. The high responsivity and the reasonable response time of approximately 100 μs make our phototransistor promising as an effective optical power monitor in Si photonic circuits.

The rapid progress in Si photonics over the past decade has made it possible to fabricate large-scale photonic circuits on a Si wafer, which are being widely used as optical interconnection devices in datacenters[1]. In addition to optical fiber communication, many applications such as deep learning[2–4], quantum computing[5–8], and sensing[9,10] are emerging for Si programmable photonic circuits[11], in which numerous optical phase shifters that can control the optical phase of light propagating in a Si waveguide with an electrical signal are integrated. To reconfigure photonic circuits for a specific purpose, it is crucial to accurately set phase values of all phase shifters. Since an initial phase error of a phase shifter is unavoidable owing to variations in device fabrication, accurate initialization and setting a target phase value are critical issues for programmable photonic circuits.

Various methods have been proposed to configure phase shifters by monitoring outputs of a photonic circuit;[12,13] however, the simplest and most reliable method is to integrate numerous optical power monitors in a photonic circuit[14–16]. For a non-invasive optical power monitoring in a waveguide, monitoring the change in the conductivity of the waveguide caused by optical absorption through capacitive coupling is one of the promising methods[17]. However, this method requires a relatively high optical input power and phase-sensitive detection with additional electronics for achieving high sensitivity. Ge photodetectors (PDs) widely used in Si photonic circuits are another candidate for use as an optical power monitor;[18] however, the dark current of a Ge PD is rather high, thereby not suitable for detecting very weak light[19,20]. Moreover, an additional photonic circuit for

[1]Department of Electrical Engineering and Information Systems, The University of Tokyo, 7-3-1 Hongo, Bunkyo-ku, Tokyo 113-8656, Japan. [2]STMicroelectronics, 850 Rue Jean Monnet, 38920 Crolles, France. ✉e-mail: takenaka@mosfet.t.u-tokyo.ac.jp

**Fig. 1 | Schematic and images of a fabricated waveguide-coupled photo-transistor. a** 3D view of the phototransistor consisting of a 30 nm-thick InGaAs membrane on the Si waveguide back gate with an $Al_2O_3$ gate dielectric. The Ni/Au metal source and drain are formed on the InGaAs channel. **b** Plan-view microscopy images of the fabricated device. An optical signal at a 1.3 μm wavelength is coupled to the Si waveguide through the grating coupler. The gate length and width are 2 μm and 30 μm, respectively.

tapping optical power and a transimpedance amplifier are required for detection, and as the number of Ge PDs increases, the complexity of photonic and electrical circuits also increases[21]. Therefore, a high-responsivity optical power monitor that can be integrated with Si programmable photonic circuits in a simple manner is essential for reconfiguring their functionality accurately.

Phototransistors based on bipolar junction transistors (BJTs)[22–26], junction field-effect transistors (JFETs)[27,28], and metal-oxide-semiconductor field-effect transistors (MOSFETs)[29–42] can have high photosensitivity owing to the amplification of photocurrent. Although BJT-based phototransistors have been developed so far for high-speed optical communication, their responsivity is typically below $10^3$ A/W because the gain for the photocurrent cannot exceed the gain for the corrector current to the base current. In contrast, MOSFET-based phototransistors called photoFETs can achieve high responsivity. A responsivity exceeding $10^6$ A/W has been reported for surface-illuminated photoFETs. However, a waveguide-coupled photoFET requires a gate electrode near the optical waveguide, which increases optical loss due to optical absorption by the gate metal[29–31]. To prevent optical absorption by the gate metal, the gate electrode should be placed away from the waveguide, losing effective gating[42,43]. As a result, high responsivity has not yet been demonstrated for waveguide-coupled photoFETs.

In this work, we present an ultrahigh-responsivity waveguide-coupled phototransistor with an InGaAs ultrathin membrane bonded onto a Si waveguide. We use the Si waveguide as a gate electrode, enabling the effective control of transistor current without optical loss, which can resolve the issue of the current waveguide-coupled photoFETs.

## Results

### Device structure and fabrication

We propose to use a Si waveguide as a back gate for a waveguide-coupled photoFET. As shown in Fig. 1a, an InGaAs ultrathin membrane is bonded onto a Si waveguide with $Al_2O_3$ gate dielectric. A gate voltage

can be applied to the InGaAs channel through the p-type Si waveguide, allowing us to eliminate the gate metal that is required for typical waveguide-coupled photoFETs. The effective control of the transistor current flowing in the InGaAs ultrathin channel is achieved while avoiding optical loss due to optical absorption by a gate metal. As a result, we can realize a highly sensitive waveguide-coupled photo-transistor. In addition to the superior electrostatic control, with the InGaAs ultrathin membrane, a tapered structure[44] is no longer required; such a structure requires a complicated fabrication process such as crystal regrowth to suppress the optical reflection at the edge of the Si hybrid waveguide[45,46]. As a result, the waveguide photo-transistor shown in Fig. 1a can be realized with a simple fabrication process (see Methods and Supplementary Section I). In particular, a buried p-type Si gate waveguide can be fabricated by the standard complementary metal-oxide-semiconductor (CMOS) processes such as ion implantation and chemical mechanical polishing. Figure 1b shows a plan-view microscopy image of the fabricated photoFET. A 30 nm-thick p-type InGaAs layer lattice-matched to InP was bonded onto the Si waveguide with 10 nm-thick $Al_2O_3$ layer. Since the $Al_2O_3$ layer works as a gate dielectric, the electron inversion layer is formed when a positive gate voltage is applied through the Si waveguide, making the transistor turn on. Since the Schottky contact for holes is formed between the p-type InGaAs and most metals owing to the Fermi level pinning[47], the metal source and drain were formed by simply depositing Ni/Au electrodes on the InGaAs layer. The low Schottky barrier height for electrons of the Ni/Au metal S/D enables the normal n-channel transistor operation. The channel length $L$ of the transistor, defined as the gap between the source and drain metals, was 2 μm. The length of the InGaAs absorber, corresponding to the width of the phototransistor $W$, was designed to be 30 μm. The Ni/Au contact to the Si waveguide back gate was formed on the heavily doped Si slab (see Supplementary Section I). The Si wave-guide was designed to obtain a single-mode operation at a 1.3 μm wavelength. A light signal at a 1.3 μm wavelength was injected into the fundamental transverse-electric (TE) mode of the Si waveguide through a grating coupler and absorbed at the 30 nm-thick InGaAs layer.

### Transistor characteristics

First, the characteristics as a transistor without light injection were investigated (see Methods and Supplementary Section II). Figure 2a shows the drain current ($I_d$) – gate voltage ($V_g$) characteristics when the drain voltages ($V_d$) were 0.05 V and 0.5 V. By applying a gate voltage to the Si waveguide back gate, we obtained a drain current on/off ratio of ~$10^4$, suggesting effective gate control for the transistor channel. Figure 2b shows the $I_d$–$V_d$ characteristics when $V_g$ was swept from −1 V to 1 V. The saturation of $I_d$ with respect to $V_d$ was observed, indicating a well-behaved transistor operation. We also prepared the InGaAs transistor for mobility measurement (see Supplementary Section III). The field-effect mobility was estimated to be 608 cm²/V·s from the $I_d$–$V_g$ characteristics. The high electron mobility contributes to the high responsivity and the reasonable response time of the proposed InGaAs photoFET as discussed later. Note that the electron mobility in the inversion channel is lower than that in lightly doped InGaAs bulk[48] due to the scattering at the MOS interface. The typical electron mobility in the inversion InGaAs channel is 1000–2000 cm²/V·s[47]. Therefore, there is room for improvement in the electron mobility of our device through process optimization.

### Photoresponses

Next, the characteristics of the photoFET were evaluated with light injection (See Methods). Figure 3a shows the $I_d$–$V_d$ characteristics at $V_g$ of 1 V when the continuous-wave light signal at a 1305 nm wavelength was injected at various input powers. Note that the input power is defined as the intensity coupled to the photoFET by excluding the

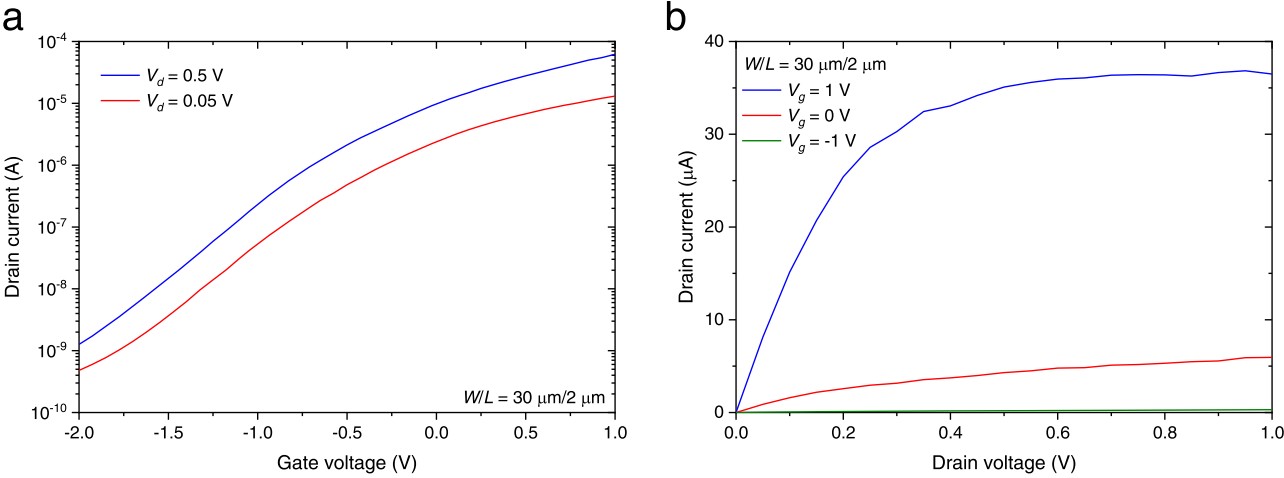

**Fig. 2 | Electrical characteristics of a waveguide-coupled InGaAs phototransistor with no light injection. a** $I_d$–$V_g$ characteristics with $V_d$ of 0.05 V and 0.5 V. **b** $I_d$–$V_d$ characteristics with $V_g$ of −1 V, 0 V, and 1 V.

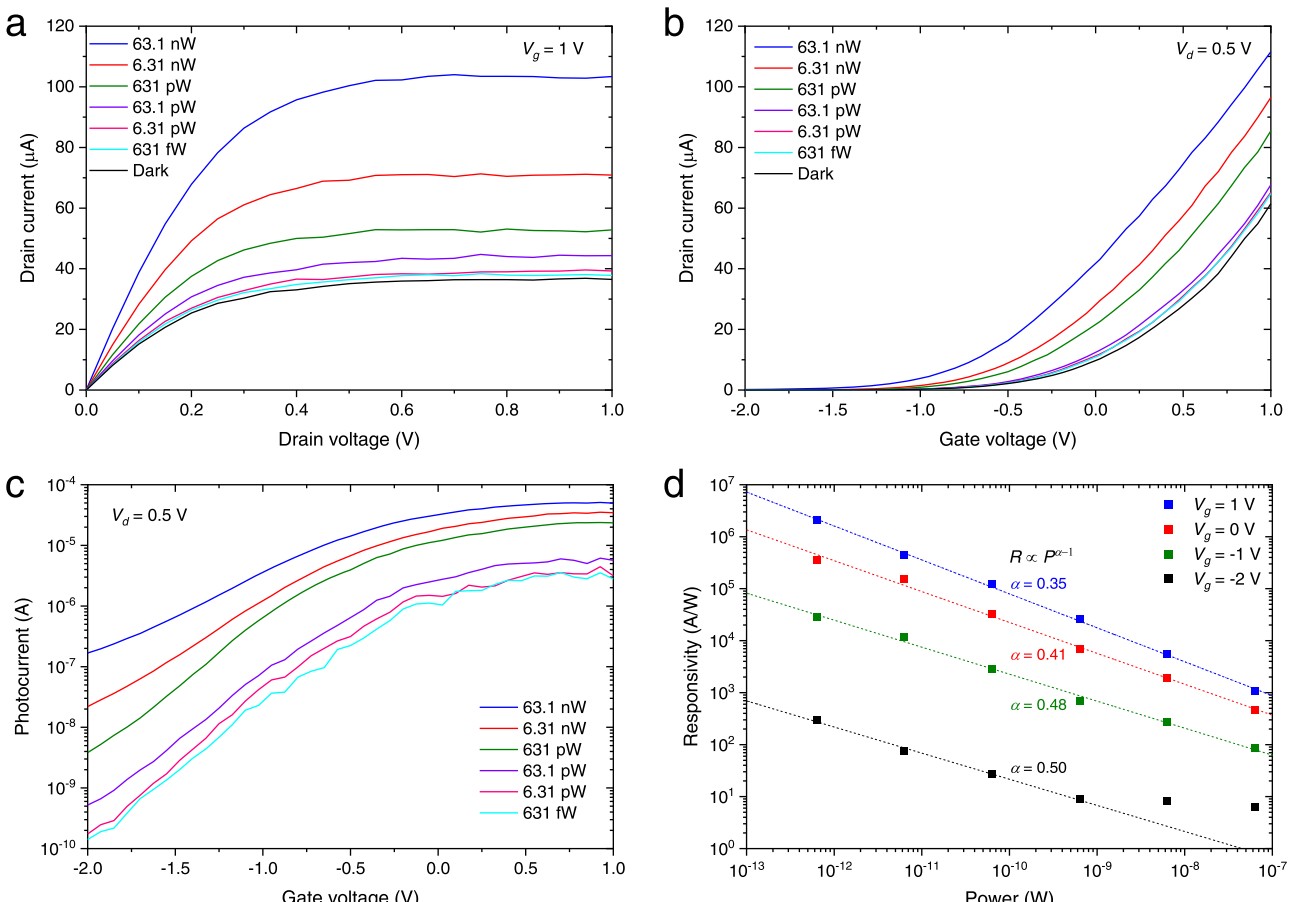

**Fig. 3 | Photoresponses of a waveguide-coupled InGaAs phototransistor with light injection of a 1.3 μm wavelength. a** $I_d$–$V_d$ characteristics with $V_g$ of 1 V. **b** $I_d$–$V_g$ characteristics in the linear scale with $V_d$ of 0.5 V. **c** Photocurrent as a function of $V_g$ with $V_d$ of 0.5 V. **d** Relationship between responsivity and input power at various gate voltages.

insertion loss of the grating coupler and the Si waveguide (see Supplementary Section IV). The total insertion loss from the grating coupler to the PD is 7.52 dB. The additional insertion loss caused by the photodetector integration can be <0.05 dB owing to the InGaAs ultrathin channel[44]. Because of the photocurrent amplification by the transistor operation, a photocurrent significantly greater than the dark

current was observed even for a very weak light of 631 fW. Figure 3b shows the $I_d$–$V_g$ characteristics in the linear scale at various input powers when $V_d$ was 0.5 V. It was clearly observed that the threshold voltage of the transistor was shifted by the photogating effect through hole accumulation in the InGaAs channel[37,39,49]. The photocurrent calculated by subtracting the dark current from the drain current with

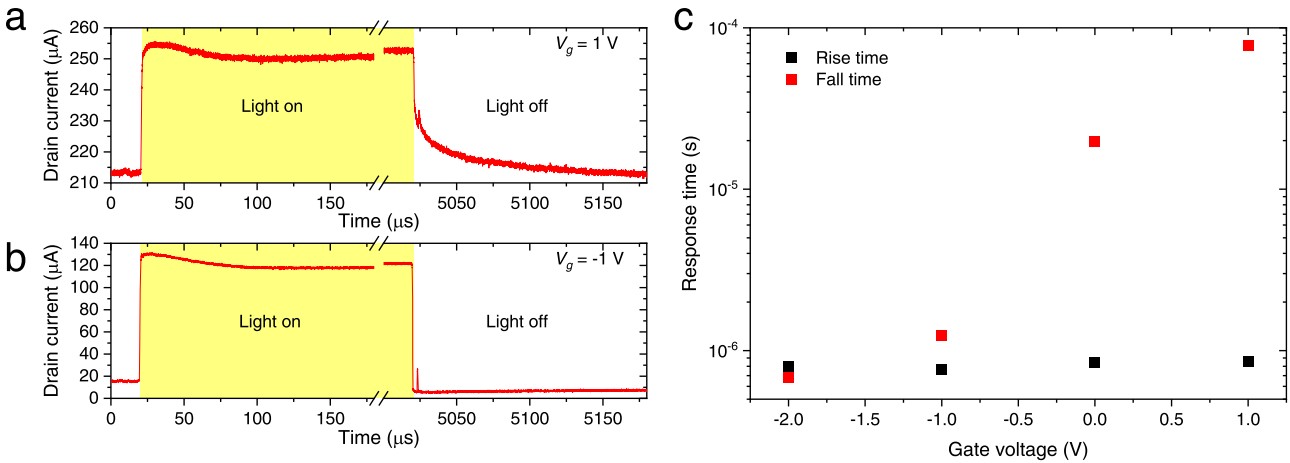

**Fig. 4 | Time response of a waveguide-coupled InGaAs phototransistor. a** Time response of $I_d$ with $V_g$ of 1 V. **b** Time response of $I_d$ with $V_g$ of −1 V. **c** Rise time and fall time as a function of $V_g$.

light irradiation is shown in Fig. 3c. A large photocurrent was obtained through the photogating effect when the transistor was in the on state. When the transistor was in the off state, the photoconductive effect might contribute to the amplification of the photocurrent as well as the photogating effect. Figure 3d shows the responsivity $R$ as a function of the input optical power $P$ at various $V_g$, where $R$ is proportional to $P^{\alpha-1}$. When $\alpha = 1$, the photoconductive effect is dominant, whereas when $0 < \alpha < 1$, the photogating effect is dominant[37,49–51]. It is found that α decreases as $V_g$ increases, suggesting that the photogating effect is more dominant in the on state. The photocurrent component attributable to the photogating effect can be expressed by the product of the transconductance and the threshold voltage shift. Since the transconductance increases as $V_g$ increases, the photocurrent amplified by the photogating effect becomes very large when the phototransistor is in the on state. As a result, when $V_g$ was 1 V, an extremely large responsivity of ~1 × 10⁶ A/W was obtained at an incident power between 631 fW and 6.31 pW. As discussed later, the absorption of the 30 μm-long phototransistor is estimated to be 6 dB. Hence, the estimated intrinsic responsivity is ~0.75 A/W at a 1.3 μm wavelength. As a result, the gain of the photocurrent is ~1.3 times greater than the responsivity. Note that the effective refractive index of the guided mode is changed by bonding the InGaAs membrane. However, this change is steady and not an issue for configuring a photonic circuit. When a gate voltage is applied, the carrier accumulation at the MOS interface may affect the effective refractive index and absorption[52]. However, since the device length is 30 μm or potentially much less than 30 μm as described later, the impact of the carrier accumulation is negligible. The leakage current flowing through the gate dielectric is very small when a gate voltage is 1 V. Therefore, there is no impact of the leakage current on our results. The $I_d$–$V_g$ curve showed small hysteresis due to the electron traps in the InGaAs/Al₂O₃ interface[53]. Since the electron traps result in a positive shift in the threshold voltage that is opposite to the photogating effect, the hysteresis does not affect our results.

### Time responses

The time response of the photoFET was also evaluated by injecting a modulated optical signal (See Methods). The rise time and fall time of the optical signal generated by direct modulation of a tunable laser were ~400 ns and 100 ns, respectively (see Supplementary Section V). Figure 4a shows the time response of $I_d$ when $V_d = 0.5$ V and $V_g = 1$ V. The rise time of $I_d$ ($\tau_R$) was ~1 μs, whereas the fall time ($\tau_F$) was ~100 μs, much longer than $\tau_R$. In contrast, when $V_g$ was −1 V, $\tau_R$ did not change significantly, whereas $\tau_F$ became much shorter (Fig. 4b). The gate voltage dependence of $\tau_R$ and $\tau_F$ is shown in Fig. 4c. Although $\tau_R$ was

almost independent of $V_g$, $\tau_F$ increased exponentially with $V_g$. When the input light was turned off, the photocurrent induced by the photogating effect decreased owing to the reduction in the number of accumulated holes in the InGaAs channel through carrier recombination. As discussed for Fig. 3d, the photogating effect becomes dominant as $V_g$ increases. As a result, the hole lifetime makes $\tau_F$ long. The hole lifetime of ~100 μs is two orders of magnitude larger than that in InGaAs bulk[54] since the electric field in the depletion layer of the InGaAs channel separates electrons and holes, making the hole lifetime long. In contrast, when $V_g$ is negative, there is no depletion layer. Therefore, the hole lifetime becomes close to that in InGaAs bulk[54], resulting in a short $\tau_F$. Note that the obtained temporal response of ~100 μs is sufficiently fast for an optical power monitor in a photonic circuit. For example, the splitting ratio of a Mach-Zehnder interferometer can be calibrated by monitoring the optical powers of two output ports.

### Discussion

Figure 5 is the benchmark for phototransistors that shows the relationship between responsivity and response time. Owing to the large optical absorption in the waveguide structure, the proposed waveguide-coupled photoFET exhibits more than three orders of magnitude greater responsivity than the surface-illuminated InGaAs photoFET[32]. Note that the InGaAs photoFET in Ref. 32 was fabricated on a flexible substrate and its responsivity can potentially be improved using a rigid substrate. PhotoFETs based on 2D materials[35,37,38,42] show a high responsivity of 10³–10⁷ A/W. However, their response time is on the order of 1 s because of the long carrier lifetime of trap states, which is more suitable for image and gas sensor applications. Phototransistors based on BJTs or JFETs exhibit a response time of <1 ns, while their responsivity is <10² A/W, which is suitable for high-speed optical communication applications. In contrast, using the InGaAs ultrathin channel and the Si waveguide gate, our device exhibits the highest responsivity of 10⁵–10⁶ A/W among the waveguide-coupled phototransistors with a sub-millisecond response time, which makes it very suitable for an optical power monitor in Si photonic circuits. The photocurrent induced by the photogating effect ($I_{PG}$) can be expressed as

$$I_{PG} = g_m \triangle V_{th} = \frac{\triangle Q}{\tau_{tr}} = \frac{g^* \tau_p}{\tau_{tr}}, \quad (1)$$

where $g_m$ is the transconductance, $\triangle V_{th}$ is the threshold voltage shift by the photogating effect, $\triangle Q$ is the total charge of the photogenerated holes, $\tau_{tr}$ is the carrier transit time of an electron[49], $g^*$ is the

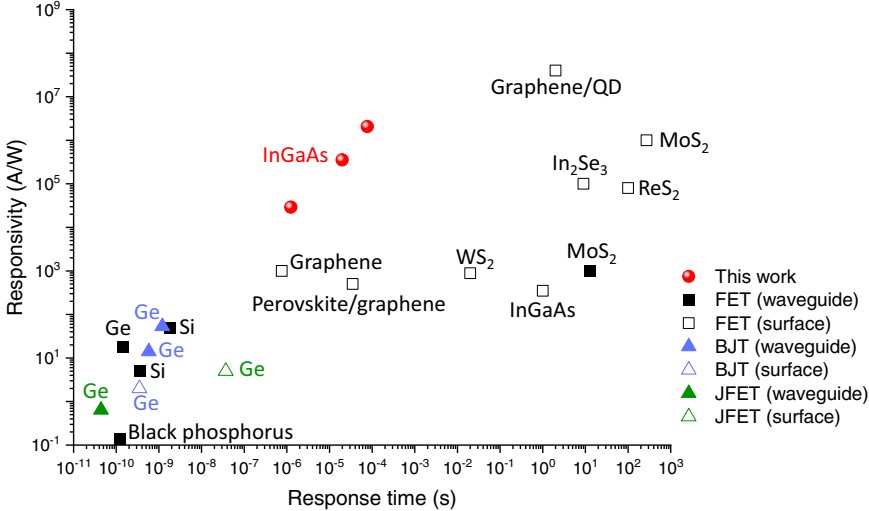

**Fig. 5 | Benchmark of responsivity and response time for various photo-transistors based on metal-oxide-semiconductor field-effect transistors (MOSFETs), bipolar junction transistors (BJTs), and junction field-effect transistors (JFETs).** Comparison of responsivity and response time demonstrated for MOSFET-based phototransistors (waveguide-coupled: Ge[31], Si[43], black phosphorus[29] and MoS$_2$[42], surface-illuminated: InGaAs[32], graphene/QD[34], perovskite/graphene[40], graphene[39], MoS$_2$[35], WS$_2$[41], In$_2$Se$_3$[37], and ReS$_2$[38]), BJT-based phototransistors (waveguide-coupled: Ge[22,24], surface-illuminated: Ge[25]), and JFET-based phototransistors (waveguide-coupled: Ge[27], surface-illuminated: Ge[28]).

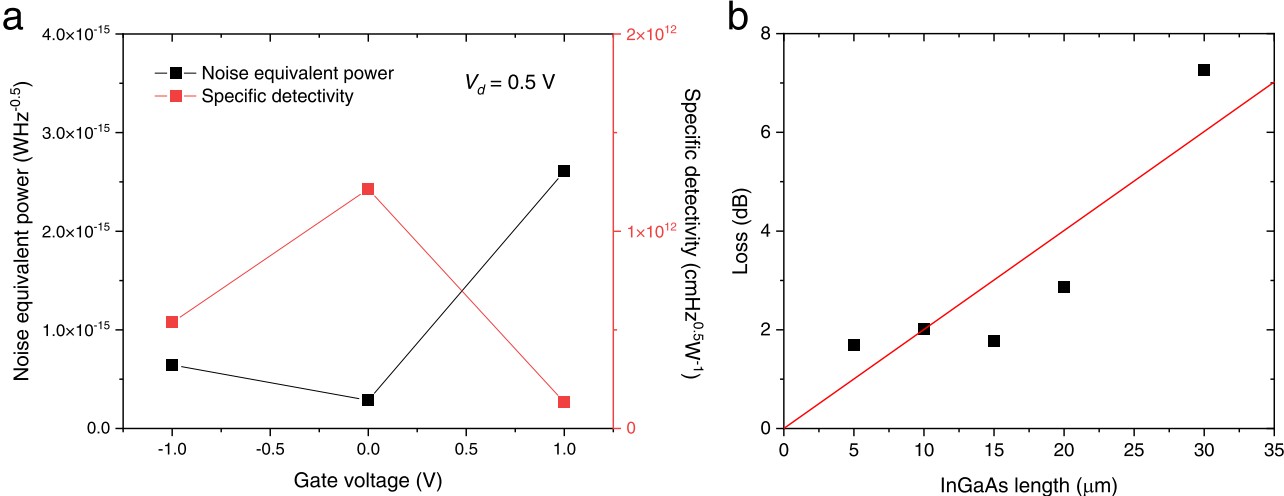

**Fig. 6 | Evaluation of noise equivalent power, specific detectivity, and insertion loss. a** Measured noise equivalent power and specific detectivity of waveguide-coupled InGaAs phototransistor. The noise equivalent power was extracted from the measured noise power density spectrum with varied gate voltages when $V_d$ was 0.5 V. **b** Insertion loss of the phototransistor as a function of InGaAs length. By measuring the optical transmission of the phototransistor with varied device lengths, the insertion loss was evaluated to be 0.20 dB/μm, suggesting that the insertion loss can be <0.1 dB with 0.5 μm-long device.

generation rate of photoexcitation, and $\tau_p$ is the carrier lifetime of holes. The proposed device has a larger $\triangle Q$ than the surface-illuminated InGaAs photoFETs because of the large optical absorption in the waveguide configuration. Since the carrier transit time $\tau_{tr}$ can be small owing to the high electron mobility in the InGaAs channel, we can increase $I_{PG}$ without increasing $\tau_p$. This is the reason why the proposed InGaAs photoFET can achieve a shorter response time than the phototransistors based on 2D materials with comparable responsivity. Since the electron mobility can be improved by process optimization due to the high electron mobility of InGaAs[47], there is room for improvement in the responsivity without the expense of the response time.

The noise equivalent power (NEP) and specific detectivity were also evaluated from the measured noise power density spectrum when $V_d$ was 0.5 V (see Supplementary Section VI). As shown in

Fig. 6a, the measured NEP took a minimum value when $V_g$ was 0 V owing to the balance between the dark current and responsivity. As a result, the specific detectivity exceeded $1 \times 10^{12}$ cm Hz$^{1/2}$ W$^{-1}$, which is ~100 times greater than that of a Ge photodetector[55]. Hence, our phototransistor exhibits high sensitivity as well as an ultrahigh responsivity.

The insertion loss of the InGaAs photodetector with varied lengths was evaluated as shown in Fig. 6b, taking into account the propagation loss of the Si waveguide and the coupling loss of the grating coupler. As expected, the insertion loss in the log scale was proportional to the InGaAs length. Since the insertion loss per unit length is 0.20 dB/μm, we expect that the insertion loss can be <0.1 dB with the 0.5 μm-long phototransistor. As the device length decreases, the dark current also decreases. In addition, the total number of holes for achieving the same threshold voltage shift also decreases since the

volume of the channel decreases with the device length. Thus, the responsivity of the scaled phototransistor is expected not to degrade markedly. Because of the high responsivity of the phototransistor, we can use it as an optical power monitor even with the device length of much smaller than 0.5 μm. Therefore, the proposed waveguide-coupled InGaAs phototransistor can potentially be used as a transparent optical power monitor for a Si waveguide.

In conclusion, we successfully demonstrated the waveguide-coupled photoFET by bonding the InGaAs ultrathin membrane onto the Si waveguide. The InGaAs ultrathin channel enhanced the photo-gating effect through the Si waveguide gating, resulting in the highest responsivity among the waveguide-coupled phototransistors. The high responsivity and the response time of our phototransistor are sufficient for optical power monitoring and are close to the performance of a commercially available optical power monitor instrument. Owing to the high responsivity, the proposed phototransistor can be a transparent in-line optical power monitor for a Si waveguide realized by reducing the length of the InGaAs absorber to <1 μm. If 10 dB insertion loss is acceptable, up to 100 photoFETs can be cascaded, enabling a $100 \times 100$ Si programmable photonic circuit. When the photoFET is biased with $V_g = 1$ V and $V_d = 0.5$ V, the transistor current is ~40 μA, meaning that the power consumption of the single photoFET is ~20 μW. In the case of a $100 \times 100$ Si programmable photonic circuit where ~10,000 monitor photoFETs should be required, the total power consumption of the photoFETs is ~200 mW. This power consumption can be reduced further by reducing the device length. Therefore, the power consumption does not limit the device density. Hence, the waveguide-coupled InGaAs photoFET with a Si waveguide gate can be used as an effective optical power monitor in Si programmable photonic circuits for communication, computing, and sensing applications.

## Methods

The InGaAs photoFET was fabricated as follows (see Supplementary Section I for the detailed fabrication procedure). A 30 nm-thick p-type $In_{0.53}Ga_{0.47}As$ membrane was bonded onto the Si waveguide with 10 nm-thick $Al_2O_3$ bonding interface. After the patterning of the InGaAs membrane, Ni/Au contact pads were formed as the source, drain, and gate by lift-off. The electrical characteristics of the photoFET were measured using a semiconductor parameter analyzer (Agilent Technologies, 4156 C). The photoresponse of the photoFET was measured at a wavelength of 1305 nm. A tunable laser (Santec, TSL-510) was used as a fiber-coupled light source. The input power was tuned using a variable optical attenuator (Anritsu, MN9605C). The polarization of the input light was adjusted to the TE mode of the Si waveguide by an in-line polarization controller. The input light was coupled from a single-mode fiber to the Si waveguide through the grating coupler. For the time response measurement, the tunable laser was modulated using an electrical waveform generator (Agilent, 33522B). The electrical waveform was recorded using a waveform generator/fast measurement unit (Agilent, B1530A) of a semiconductor device analyzer (Agilent Technologies, B1500A).

## Data availability

The data that support the findings of this study are available from the corresponding authors on reasonable request.

## Code availability

The code that used for the simulation is available at https://doi.org/10.5281/zenodo.7281340.

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

## Acknowledgements

This work was partly based on results obtained from projects (JPNP14004, JPNP16007 received by M.T.) commissioned by the New Energy and Industrial Technology Development Organization (NEDO) and partly supported by JST, CREST (JPMJCR2004 received by M.T.). Part of this work was conducted at Takeda Sentanchi super cleanroom, The University of Tokyo, supported by "Nanotechnology Platform Pro-gram" of the Ministry of Education, Culture, Sports, Science and Tech-nology (MEXT), Japan, Grant Number JPMXP09F20UT0021. The authors thank Dr. H. Yagi, Dr. Y. Itoh, and Dr. H. Mori of Sumitomo Electric for providing InP epitaxial wafers.

## Author contributions

T.O. contributed to fabrication, measurement, and paper preparation. T.A. contributed to measurement. Y.M. contributed to simulation. K.S. and S.O. contributed to the fabrication and analysis. S.M. and F.B. con-tributed to design and fabrication of the Si waveguide. K.T. and S.T. contributed to the overall discussion. M.T. contributed to the idea, dis-cussion, and paper revision and also provided high-level project supervision.

## Competing interests

The authors declare no competing interests.
