## [Peer Review file · Nature Communications]

We would like to thank all the reviewers for kindly reviewing our paper and highly appreciate their insightful comments on our research. We have revised our manuscript appropriately in response to the reviewers' comments from the submission to Nature communications. We have added two authors who have contributed to the paper revision. We believe that the revisions have improved our paper, appropriate to the publication in Nature communications. The revised parts are highlighted in red color and the newly added/reconstructed parts are highlighted with blue color. Please find the details in the authors' response to each specific comment.

Sincerely,

Mitsuru Takenaka, Professor
Department of Electrical Engineering and Information Systems,
The University of Tokyo,
7-3-1 Hongo, Bunkyo-ku, Tokyo 113-8656, Japan
E-mail : takenaka@mosfet.t.u-tokyo.ac.jp
Phone : +81-3-5841-6733
Fax : +81-3-5841-8564

Reviewers' Comments:

#####

Reviewer #2 (Remarks to the Author):

In this work, Ochiai and colleagues report on the implementation and characterization of infrared photodetectors coupled to Si waveguides for Si photonic circuit applications. They integrate a phototransistor architecture that uses the Si waveguide as a backgate, coated with a thin layer of alumina (insulator) and a thin layer of InGaAs as the photoactive material. Using this arrangement, the authors achieve high responsivity ($R \sim 1E6$ A/W) at 1.3 μ m, a sub ms time response, and are able to detect dim light (100s of fW). As it stands, this set of results don't seem sufficient to recommend publication in Nature Photonics. This could change depending on the outcome of the missing characterization as detailed below:

The authors offer a nice introduction and motivation to the need of integrated photodetectors in Silicon photonic circuits. They specifically mention the case-application of phase-shifter calibration. They do also offer an overview of existing technologies to address this challenge, such as Ge or SiGe photodetectors – which, they argue, suffer from elevated dark currents.

Comment #1

The next questions are thus key to evaluate the performance and impact of the proposed solution:

- Time response: a proper calibration of said phase-shifters would require the integrated photodetectors to be able to resolve the waveguided signals with sufficient temporal resolution. Since one of the main applications of Si photonics is high speed, is the achieved temporal response (100s us) sufficient to address signals in the MHz and GHz regime?

Response:

Thank you for this comment. An optical power monitor discussed in this paper is expected to be used to configure an optical circuit. For example, the splitting ratio of a Mach-Zehnder interferometer can be calibrated by monitoring the optical powers of two output ports. For this purpose, we don't have to detect a high-speed optical signal. Hence, the current temporal response of around 100 μ s is sufficient for an optical power monitor in a photonic circuit. We have revised our manuscript as follows.

Page 8:

Original:

...the accumulated holes are more easily swept out from the channel through quantum tunneling, resulting in a short τ_F .

Revised:

...the accumulated holes are more easily swept out from the channel through quantum tunneling, resulting in a short τ_F . **Note that the obtained temporal response of approximately 100 μ s is sufficiently fast for an optical power monitor in a photonic circuit. For example, the splitting ratio of a Mach-Zehnder interferometer can be calibrated by monitoring the optical powers of two output ports.**

Comment #2

- Dark current: the authors make a big deal about the responsivity, as it allows to 'amplify' the signal to be detected. However, they do not offer a direct comparison with other works in terms of dark current and noise.

As such:

- This reviewer recommends that the authors perform a direct comparison in the gain * dark current * bandwidth space.

- The dark current, but also the measured noise and measured specific detectivity should be provided in order to enable a fair comparison with existing alternatives. One could for example have

extraordinarily high responsivities but, if they come associated to excessive noise, would be impractical.

Response:

Thank you for the insightful comment. Since a phototransistor exhibits a high gain when it is on, it might be hard to make a fair comparison with respect to dark current. It might also be difficult to define a gain clearly because an intrinsic quantum efficiency cannot be measured directly. Moreover, most papers regarding waveguide phototransistors do not report the specific detectivity. Therefore, in this paper, we use responsivity as a benchmark. However, as the reviewer pointed out, to discuss noise further, we measured the noise equivalent power (NEP) and specific detectivity of our device. We found that the noise equivalent power took a minimum value when $V_g = 0$ V, and the measured detectivity exceeds 1×10^{12} cm Hz^{1/2} W⁻¹, which is approximately 100 times greater than that of a Ge photodetector⁵⁵. To clarify this point, we added the discussion about the NEP and specific detectivity in the manuscript and added the details in the measurements in supplementary.

Reference:

55. Lin, Y., Lee, K. H., Son, B. & Tan, C. S. Low-power and high-detectivity Ge photodiodes by in-situ heavy As doping during Ge-on-Si seed layer growth. *Opt. Express* **29**, 2940–2952 (2021)

Page 9:

The noise equivalent power (NEP) and specific detectivity were also evaluated from the measured noise power density spectrum when V_d was 0.5 V (see Supplementary Section V). As shown in Fig. 6, the measured NEP took a minimum value when V_g was 0 V owing to the balance between the dark current and responsivity. As a result, the specific detectivity exceeded 1×10^{12} cm Hz^{1/2} W⁻¹, which is approximately 100 times greater than that of a Ge photodetector⁵⁵. Hence, our phototransistor exhibits high sensitivity as well as an ultrahigh responsivity.

Fig. 6

Section VI in Supplementary

The noise power density spectrum of an InGaAs phototransistor was evaluated by the Fourier transform of the dark current waveform measured using a waveform generator/fast measurement unit (Agilent, B1530A) of a semiconductor device analyzer (Agilent Technologies, B1500A). Figure S10 shows the measured noise power density spectrum when V_d and V_g were 0.5 V and 1.0 V, respectively. As shown in Fig. S10, the noise power density was proportional to $1/f^2$, where f is a frequency. According to the method described in Ref. 4, the noise equivalent power (NEP) was extracted by integrating the noise power density from 0.1 Hz to 10 kHz. The specific detectivity was then obtained from the NEP, where the area of the phototransistor was the product of the waveguide width (0.4 μm) and InGaAs length (30 μm).

Fig. S10. Measured noise power density of waveguide-coupled InGaAs phototransistor. V_d and V_g was 0.5 V and 1.0 V, respectively.

Reference:

4. Weng, W. Y., Chang, S. J., Hsu, C. L. & Hsueh, T. J. A ZnO-nanowire phototransistor prepared on glass substrates. *ACS Appl. Mater. Interfaces* **3**, 162–166 (2011).

Comment #3

- Integration: the authors argue that their proposed approach offers ease of integration. However, they employ a buried p⁺ Si gate. Can the authors comment on that?

Response:

Thank you for this comment. A p⁺-Si waveguide can be easily fabricated by ion implantation and following annealing, which is a quite regular process. After deposition of the SiO₂ cladding, surface planarization is carried out by chemical mechanical polishing, which is also very standard in the CMOS process. Indeed, the Si waveguide in this paper was prepared using the CMOS fab of STMicroelectronics. There is no fundamental issue to prepare a buried p⁺-Si gate. To emphasize this discussion, we have revised the manuscript as below.

Page 4:

Original:

...As a result, the waveguide phototransistor shown in Fig. 1a can be realized with a simple fabrication process (see Methods and Supplementary Section I).

Revised:

...As a result, the waveguide phototransistor shown in Fig. 1a can be realized with a simple fabrication process (see Methods and Supplementary Section I). **In particular, a buried p-type Si gate waveguide can be fabricated by the standard complementary metal-oxide-semiconductor (CMOS) processes such as ion implantation and chemical mechanical polishing.**

Comment #4

- Sensing approach: the proposed approach (using Si/pSi as a gate, coated with Al₂O₃ to detect evanescent fields in the InGaAs slab) is elegant; but have the authors considered potential impact on the performance of the Si waveguides? Would the guided modes experience increased dispersion in view of the modified environment (permittivity, charge accumulation and electrostatics due to gating)?

Response:

Thank you for the important comment. As the reviewer pointed out, the effective refractive index of the guided mode is changed by bonding the InGaAs membrane. However, this change is steady and not an issue for configuring a photonic circuit. When a gate voltage is applied, the carrier accumulation at the MOS interface may affect the effective refractive index and absorption⁵². However, since the device length is 30 μm or potentially much less than 30 μm, the impact of the charge accumulation is negligible. To clarify this point, we have revised the manuscript as below.

Page 7:

Original:

...an extremely large responsivity of 2.1×10^6 A/W was obtained at an incident power of 631 fW.

Revised: (the green part is the revision for Comment #10)

.....an extremely large responsivity of 2.1×10^6 A/W was obtained at an incident power of 631 fW. **As discussed later, the absorption of the 30-μm-long phototransistor is estimated to be 6 dB. Hence, the estimated intrinsic responsivity is approximately 0.75 A/W at a 1.3 μm wavelength. As a result, the gain of the photocurrent is approximately 1.3 times greater than the responsivity. Note that the effective refractive index of the guided mode is changed by bonding the InGaAs membrane. However, this change is steady and not an issue for configuring a photonic circuit. When a gate voltage is applied, the carrier accumulation at the MOS interface may affect the effective refractive index and absorption⁵². However, since the device length is 30 μm or potentially much less than 30 μm as described later, the impact of the carrier accumulation is negligible.**

Reference:

52. Han, J.-H. *et al.* Efficient low-loss InGaAsP/Si hybrid MOS optical modulator. *Nat. Photonics* **11**, 486–490 (2017)

Comment #5

- On the same note, it would be helpful to see the E-field distribution upon gating using some modelling.

Response:

Thank you for this suggestion. We have added the electrical field distribution in Supplementary.

Page 5:

Original:

First, the characteristics as a transistor without light injection were investigated (see Method).

Revised:

First, the characteristics as a transistor without light injection were investigated (see Method **and Supplementary Section II**).

Section II in Supplementary

The operation principle of a phototransistor with the metal source and drain is depicted in Fig. S3. Since n-type transistor is considered here, the Schottky contact with large barrier height for holes are assumed. When no light is injected, the transistor is off, resulting in a low drain current. When the transistor channel is irradiated by light, photo-generated holes accumulate in the channel, pushing the conduction band and valence band down. As a result, the transistor turns on, and more drain current flows through the channel. In this way, the photocurrent is amplified through the change in the transistor conduction.

Fig. S3. **Operation principle of a phototransistor with metal source and drain.**

The band diagram of the proposed waveguide-coupled InGaAs phototransistor was simulated using Ansys Lumerical DEVICE when V_d and V_g are 0.5 V and 0 V, respectively, as shown in Fig. S4. Here, the Schottky contact with a barrier height of 0.1 eV for electrons was assumed as the metal source and drain. Here, an n-type InGaAs layer with a doping density of $5 \times 10^{16} \text{ cm}^{-3}$ was assumed to represent the negative threshold voltage observed in the experiments. As shown in Fig. S4, the channel under the gate has a potential barrier for electrons that is modulated by light injection. The distributions of the electro-static potential and electric field are also shown in Fig. S5. Note that the electrical contact to the Si waveguide was set at the bottom of the Si layer due to the limitation of the simulation.

Fig. S4. **Band diagram of waveguide-coupled InGaAs phototransistor across the channel direction.**

Fig. S5. **Distributions of the electric-static potential and electric field in waveguide-coupled InGaAs phototransistor.** **a**, Electro-static potential when V_d and V_g are 0.5 V and 0 V , respectively. **b**, Electric field when V_d and V_g are 0.5 V and 0 V , respectively.

Comment #6

- Line 53: ultrahigh-sensitivity is not the correct term with the current data set. The authors could mention high responsivity (not ultrahigh, or actually offer a quantitative comparison with respect prior art) but not sensitivity, since they do not actually characterize the detectivity of the devices.

Response:

Thank you for insightful comment. We agree that “ultrahigh-sensitivity” is not proper to represent our results. Our original intention is to say “ultrahigh-responsivity” since we benchmark our result using responsivity as shown in Fig. 5. Our device exhibits $10^3 - 10^4$ greater responsivity than other waveguide-coupled phototransistors. Thus, we can say that our device has an ultrahigh responsivity among waveguide-coupled phototransistors. To clarify this discussion, we have revised the title as below and changed “ultrahigh-sensitivity” to “ultrahigh-responsivity” in the manuscript. In addition, we have added the discussion about the noise equivalent power (NEP) and specific detectivity as shown in the response for Comment #2.

Title:

Original:

Ultrahigh-sensitivity optical power monitor for Si photonic circuits

Revised:

Ultrahigh-**responsivity waveguide-coupled** optical power monitor for Si photonic circuits **operating at near-infrared wavelengths**

Abstract:

Original:

A phototransistor is a promising candidate as an optical power monitor in Si photonic circuits since the internal gain of photocurrent enables high sensitivity.

Revised:

A phototransistor is a promising candidate as an optical power monitor in Si photonic circuits since the internal gain of photocurrent enables high **responsivity.**

Page 3:

Original:

Therefore, a high-sensitivity optical power monitor that can be integrated with Si programmable photonic circuits in a simple manner is essential for reconfiguring their functionality accurately.

Revised:

Therefore, a high-**responsivity** optical power monitor that can be integrated with Si programmable photonic circuits in a simple manner is essential for reconfiguring their functionality accurately.

Page 3:

Original:

In this paper, we present an ultrahigh-sensitivity waveguide-coupled phototransistor with an InGaAs ultrathin membrane.

Revised:

In this paper, we present an ultrahigh-**responsivity** waveguide-coupled phototransistor with an InGaAs ultrathin membrane.

Comment #7

- The wavelength of interest (1.3 μm) is only first mentioned in line 93. This information is very important, and as such should be included in the abstract and maybe even in the title itself.

Response:

Thank you for this comment. In this paper, we used 1.3 μm wavelength since the library of the Si waveguide provided by STMicroelectronics is only for 1.3 μm . However, in principle, our device can work at a near-infrared wavelength from 1.3 μm to 1.6 μm typically used for optical fiber communication. To clarify this point, we have revised the title and abstract as below.

Title:

Original:

Ultrahigh-sensitivity optical power monitor for Si photonic circuits

Revised:

Ultrahigh-**responsivity waveguide-coupled** optical power monitor for Si photonic circuits **operating at near-infrared wavelengths**

Abstract:

Original:

Here, we present a waveguide-coupled phototransistor consisting of an InGaAs ultrathin channel on a Si waveguide working as a gate electrode to increase the responsivity.

Revised:

Here, we present a waveguide-coupled phototransistor operating at a 1.3 μm wavelength, which consists of an InGaAs ultrathin channel on a Si waveguide working as a gate electrode to increase the responsivity.

Comment #8

- Line 110: Defining the input power as the one excluding insertion losses makes sense to evaluate the performance of the photodetector. However, from the application/system level point of view, it would be useful to be more explicit regarding the amount of insertion losses (and the potential losses due to the photodetector integration itself)

Response:

Thank you for the insightful comment. The coupling loss from an optical fiber to the Si waveguide and the propagation loss of the Si waveguide are 7.5 dB and 2.14 dB/cm, respectively, as discussed in the supplementary Fig. S8. Since the length of the Si waveguide from the grating coupler to the PD is 95 μm , the insertion loss, which is dominated by the grating coupler loss, is 7.52 dB.

Since we use an ultrathin InGaAs membrane, the additional insertion loss caused by the photodetector integration can be less than 0.05 dB, as discussed in Ref. 44. To clarify this point, we have revised the manuscript and supplementary.

Page 6:

Original:

... Note that the input power is defined as the intensity coupled to the photoFET by excluding the insertion loss of the grating coupler and the Si waveguide (see Supplementary Section III).

Revised:

... Note that the input power is defined as the intensity coupled to the photoFET by excluding the insertion loss of the grating coupler and the Si waveguide (see Supplementary Section III). **The total insertion loss from the grating coupler to the PD is 7.52 dB. The additional insertion loss caused by the photodetector integration can be less than 0.05 dB owing to the InGaAs ultrathin channel⁴⁴.**

44. Ohno, S. *et al.* Taperless Si hybrid optical phase shifter based on a metal-oxide-semiconductor capacitor using an ultrathin InP membrane. *Opt. Express* **28**, 35663-35673 (2020).

Supplementary Section IV:

Original:

... From the results in Fig. S5b, the propagation loss of the Si waveguide and the coupling loss of the grating coupler were extracted to be 2.14 dB/cm and 7.5 dB, respectively, as shown in Fig. S5c.

Revised:

... From the results in Fig. S8b, the propagation loss of the Si waveguide and the coupling loss of the grating coupler were extracted to be 2.14 dB/cm and 7.5 dB, respectively, as shown in Fig. S8c. **Since the length of the Si waveguide from the grating coupler to the PD is 95 μm , the total insertion loss, which is dominated by the coupling loss of the grating coupler, is 7.52 dB.**

Comment #9

- The authors should offer a more mechanistic understanding on the processes leading to gain. They do already have the needed information such as rise and fall times for different gates and powers.
- A band diagram would help the reader understand the process better.

Response:

Thank you for this comment. As discussed in Fig. 3d, the origin of the gain relies on photogating caused by photo-generated holes accumulated in the InGaAs thin channel. The accumulated holes shift the threshold voltage of the transistor, resulting in the gain in photocurrent. To clarify this point, we have added the explanation of the operation principle and band diagram in Supplementary.

Please see the details in the response for Comment #5.

Comment #10

- What is the EQE for the primary photocurrent (excluding gain) and what is then the actual gain considering the measured responsivity?

Response:

Thank you for this comment. Since the photocurrent is amplified inside the device, we are not able to extract the EQE and gain directly from the measurements. The absorption of the PD itself was measured to be approximately 0.20 dB/ μm . Thus, the absorption of the 30- μm -long PD is estimated to be 6 dB. If the intrinsic quantum efficiency is assumed to 100%, the intrinsic responsivity is approximately 0.75 A/W at a 1.3 μm wavelength. Therefore, the gain is approximately 1.3 times greater than the responsivity. To clarify this point, we have revised the manuscript as below.

Page 6:

Original:

...an extremely large responsivity of 2.1×10^6 A/W was obtained at an incident power of 631 fW.

Revised: (the green part is the revision for Comment #4)

.....an extremely large responsivity of 2.1×10^6 A/W was obtained at an incident power of 631 fW. As discussed later, the absorption of the 30- μm -long phototransistor is estimated to be 6 dB. Hence, the estimated intrinsic responsivity is approximately 0.75 A/W at a 1.3 μm wavelength. As a result, the gain of the photocurrent is approximately 1.3 times greater than the responsivity. Note that the effective refractive index of the guided mode is changed by bonding the InGaAs membrane. However, this change is steady and not an issue for configuring a photonic circuit. When a gate voltage is applied, the carrier accumulation at the MOS interface may affect the effective refractive index and absorption⁵¹. However, since the device length is 30 μm or potentially much less than 30 μm as described later, the impact of the carrier accumulation is negligible.

Comment #11

- Could the authors comment on the leakage current of their proposed approach? Do these devices show any kind of hysteresis?

Response:

Thank you for this comment. The leakage current flowing through the gate dielectric is very small when a gate voltage is 1 V. Therefore, there is no impact of the leakage current on our results. An InGaAs MOSFET tends to show small hysteresis due to the electron traps at InGaAs/Al₂O₃ interface. Since the electron traps result in a positive shift in the threshold voltage (opposite to our result), we can conclude that our results are not affected by the hysteresis. We have clarified this point in the revised manuscript as below.

Page 7:

Original: (the green part is the revision for Comment #10)

...the impact of the carrier accumulation is negligible.

Revised:

...the impact of the carrier accumulation is negligible. The leakage current flowing through the gate dielectric is very small when a gate voltage is 1 V. Therefore, there is no impact of the leakage current on our results. The I_d-V_g showed small hysteresis due to the electron traps in the InGaAs/Al₂O₃ interface⁵³. Since the electron traps result in a positive shift in the threshold voltage that is opposite to

the photogating effect, the hysteresis does not affect our results.

53. Lin, J. *et al.* An investigation of capacitance-voltage hysteresis in metal/high-k/In_{0.53}Ga_{0.47}As metal-oxide-semiconductor capacitors. *J. Appl. Phys.* **114**, 144105 (2013).

Comment #12

- Line 151: the authors compare the performance of their InGaAs photodetector with reference 32 (a flexible InGaAs photodetector). This is not fair as flexible photodetectors tend to underperform compared to their crystalline rigid counterparts.

Response:

Thank you for this comment. We basically agree this comment, while the responsivity of Ref. 32 is better than that of Ref. 33 (a InGaAs phototransistor on the rigid substrate. Therefore, we compared our result with Ref. 32. We have argued that Ref. 32 reported a flexible InGaAs phototransistor for more fair comparison in the revised manuscript.

Page 8:

Original:

...Owing to the large optical absorption in the waveguide structure, the proposed waveguide-coupled photoFET exhibits more than three orders of magnitude greater responsivity than the surface-illuminated InGaAs photoFET³².

Revised:

...Owing to the large optical absorption in the waveguide structure, the proposed waveguide-coupled photoFET exhibits more than three orders of magnitude greater responsivity than the surface-illuminated InGaAs photoFET³². **Note that the InGaAs photoFET in Ref. 32 was fabricated on a flexible substrate and its responsivity can potentially be improved using a rigid substrate.**

#####

Reviewer #3 (Remarks to the Author):

The manuscript reports on the characterization of a phototransistor based on a thin InGaAs layer (30 nm) bonded on a silicon waveguide via an ultrathin Al₂O₃ layer (10nm). The transistor effect is obtained in the InGaAs layer (the channel), which is contacted at both ends (source and drain contacts), by biasing the silicon layer (gate contact). Under illumination, a shift in the characteristics Id-Vg is observed, reflecting the optical control of the phototransistor. This phototransistor shows a strong

responsivity. Its photo-response is intrinsically slow (100 μ s) but could be fast enough for monitoring the silicon optical circuits.

The manuscript is well written and presents both interesting and promising results. However, the physics involved in this device is not sufficiently explained to allow publication in Nature Photonics. Below is the list of the points that bother me the most.

Comment #1

- Line 151: the author

The operating modes of the device (with and without illumination) are not described in the manuscript. This description must be provided in the text since the proposed structure differs from a field effect transistor on several points. The most important is certainly the nature of the source and drain contacts. Being deposited on a very lightly doped InGaAs layer ($N_D = 5 \times 10^{16} \text{ cm}^{-3}$) they form Schottky contacts instead of the expected ohmic contacts, introducing fundamental differences from the FET behaviour such as non-linearities of the current injected into the channel, bipolar behaviour (electron and holes currents are injected at the drain and source contacts), populations of carriers out of thermodynamic equilibrium in the vicinity of these contacts.

Response:

Thank you for this comment. However, there seems to be a misunderstanding about the operation principle of our device. In this paper, we used a p-type InGaAs (not n-type as the reviewer considered), and formed Schottky contact for holes, meaning we formed Ohmic contacts for electrons. Therefore, our device works as a normal n-FET even with such a metal S/D⁴⁷. As a result, the operation principle relies on photogating⁴⁹ caused by photo-generated holes accumulated in the InGaAs thin channel. The accumulated holes shift the threshold voltage of the transistor, resulting in the gain in photocurrent. We have clarified this point and added the explanation of the operation principle with the band diagram in the revised manuscript and supplementary as below.

References:

47. Kim, S. H. *et al.* High Performance extremely thin body InGaAs-on-insulator metal–oxide–semiconductor field-effect transistors on Si substrates with Ni–InGaAs metal source/drain. *Appl. Phys. Express* **4**,114201 (2011).
49. Fang, H. & Hu, W. Photogating in low dimensional photodetectors. *Adv. Sci.* **4**, 1700323 (2017).

Page 4:

Original:

...Since the Schottky contact is formed between the p-type InGaAs and most metals owing to the

Fermi level pinning⁴⁷, the metal source and drain were formed by simply depositing Ni/Au electrodes on the InGaAs layer.

Revised:

...Since the Schottky contact **for holes** is formed between the p-type InGaAs and most metals owing to the Fermi level pinning⁴⁷, the metal source and drain were formed by simply depositing Ni/Au electrodes on the InGaAs layer. **The low Schottky barrier height for electrons of the Ni/Au metal S/D enables the normal n-channel transistor operation.**

Section II in Supplementary

The operation principle of a phototransistor with the metal source and drain is depicted in Fig. S3. Since n-type transistor is considered here, the Schottky contact with large barrier height for holes are assumed. When no light is injected, the transistor is off, resulting in a low drain current. When the transistor channel is irradiated by light, photo-generated holes accumulate in the channel, pushing the conduction band and valence band down. As a result, the transistor turns on, and more drain current flows through the channel. In this way, the photocurrent is amplified through the change in the transistor conduction.

Fig. S3. Operation principle of a phototransistor with metal source and drain.

The band diagram of the proposed waveguide-coupled InGaAs phototransistor was simulated using Ansys Lumerical DEVICE when V_d and V_g are 0.5 V and 0 V, respectively, as shown in Fig. S4. Here, the Schottky contact with a barrier height of 0.1 eV for electrons was assumed as the metal source and drain. Here, an n-type InGaAs layer with a doping density of $5 \times 10^{16} \text{ cm}^{-3}$ was assumed to represent

the negative threshold voltage observed in the experiments. As shown in Fig. S4, the channel under the gate has a potential barrier for electrons that is modulated by light injection. The distributions of the electro-static potential and electric field are also shown in Fig. S5. Note that the electrical contact to the Si waveguide was set at the bottom of the Si layer due to the limitation of the simulation.

Fig. S4. **Band diagram of waveguide-coupled InGaAs phototransistor across the channel direction.**

Fig. S5. **Distributions of the electric-static potential and electric field in waveguide-coupled InGaAs phototransistor.** **a**, Electro-static potential when V_d and V_g are 0.5 V and 0 V , respectively. **b**, Electric field when V_d and V_g are 0.5 V and 0 V , respectively.

Comment #2

The hypothesis of an operation close to that of a FET could explain that certain results are in complete disagreement with those of the literature:

1 / The electron mobility extracted from the Id-Vg characteristic of a test transistor ($\mu_n = 608 \text{ cm}^2/\text{V}\cdot\text{s}$) is clearly lower than that reported in the literature for an InGaAs layer as well lightly doped ($\mu_n > 4000 \text{ cm}^2/\text{V}\cdot\text{s}$ [M. Sotoodeh et al. J. Appl. Phys. 87(6) 2890 (2000)]);

Response:

Thank you for this comment. The literature [M. Sotoodeh et al. J. Appl. Phys. 87(6) 2890 (2000)] reported the electron mobility in a bulk InGaAs, which is different from the electron mobility in the inversion channel of the MOSFET that we discussed in this paper. Since the inversion electrons transport at the MOS interface, the electron mobility of the FET is lower than that in the bulk material. The electron mobility of the InGaAs MOSFET is typically $1000 - 2000 \text{ cm}^2/\text{V}\cdot\text{s}$, as reported in Ref. 47. Therefore, the mobility in this paper is not significantly lower than this typical value. We can improve the electron mobility by process optimization. We have clarified this point in the revised manuscript.

Page 5:

Original:

...The high electron mobility contributes to the high responsivity and the reasonable response time of the proposed InGaAs photoFET as discussed later.

Revised:

...The high electron mobility contributes to the high responsivity and the reasonable response time of the proposed InGaAs photoFET as discussed later. **Note that the electron mobility in the inversion channel is lower than that in lightly doped InGaAs bulk⁴⁸ due to the scattering at the MOS interface. The typical electron mobility in the inversion InGaAs channel is $1000 - 2000 \text{ cm}^2/\text{V}\cdot\text{s}$ ⁴⁷. Therefore, there is room for improvement in the electron mobility of our device through process optimization.**

References:

47. Kim, S. H. *et al.* High Performance extremely thin body InGaAs-on-insulator metal–oxide–semiconductor field-effect transistors on Si substrates with Ni–InGaAs metal source/drain. *Appl. Phys. Express* **4**,114201 (2011).

48. Sotoodeh, M., Khalid, A. H. & Rezazadeh, A. A. Empirical low-field mobility model for III–V compounds applicable in device simulation codes. *J. Appl. Phys.* **87**, 2890–2900 (2000).

Comment #3

2 / The hole lifetime extracted from the time response of the phototransistor for positive gate voltages (100 μ s) is two orders of magnitude larger than the values reported in the literature ($\sim 1 \mu$ s [A.W. Walker et al., *Appl. Phys. Lett.* 111 162107 (2017)]);

3 / On the other hand, the one attributed to a tunnel escape time corresponds fairly well to this value.

Response:

Thank you for the important comment. The literature [A.W. Walker et al., *Appl. Phys. Lett.* 111 162107 (2017)] reported the hole lifetime in an InGaAs bulk. In contrast, our devices reflect the hole lifetime in the InGaAs channel under gate bias. In particular, when $V_g = 1$ V, there is the depletion layer in the InGaAs channel with the vertical electric field. Therefore, electrons and holes are separated by this vertical field. As a result, the hole lifetime in the depleted channel is longer than in an InGaAs bulk. When V_g is negative, there is no depletion layer in the channel. Therefore, the hole lifetime becomes close to the that in bulk, as the reviewer pointed out. We have clarified this point in the revised manuscript as below.

Page 8:

Original:

...As a result, the hole lifetime makes τ_F long. In contrast, when the photoconductive effect is dominant with a negative V_g , the accumulated holes are more easily swept out from the channel through quantum tunneling, resulting in a short τ_F .

Revised:

...As a result, the hole lifetime makes τ_F long. **The hole lifetime of approximately 100 μ s is two orders of magnitude larger than that in InGaAs bulk⁵⁴ since the electric field in the depletion layer of the InGaAs channel separates electrons and holes, making the hole lifetime long. In contrast, when V_g is negative, there is no depletion layer. Therefore, the hole lifetime becomes close to that in InGaAs bulk⁵⁴, resulting in a short τ_F .**

Reference:

54. Walker, A. W. & Denhoff, M. W. Heavy and light hole minority carrier transport properties in low-doped n-InGaAs lattice matched to InP. *Appl. Phys. Lett.* **111**, 162107 (2017)

Comment #4

The maximum value of the responsivity obtained at very low incident power is questionable:
1 / At such a low incident power there is no significant difference between the photo-responses at 6.31pW and 0.631pW: Fig. 3c shows that the difference between these two photocurrents is in the noise range;

Response:

Thank you for the important comment. We agree that the photocurrent at 0.631 pW is very close to the noise. What we would like to say is that the responsivity is around 1×10^6 A/W (not 2.1×10^6 A/W) which can be achieved between 0.631 pW and 6.31 pW. To clarify this point, we have revised the manuscript.

To discuss noise, we have also added the discussion about the noise equivalent power (NEP) and specific detectivity of our device. The measured detectivity exceeds 1×10^{12} cm Hz^{1/2} W⁻¹, which is approximately 100 times greater than that of a Ge photodetector⁵⁵.

Reference:

55. Lin, Y., Lee, K. H., Son, B. & Tan, C. S. Low-power and high-detectivity Ge photodiodes by in-situ heavy As doping during Ge-on-Si seed layer growth. *Opt. Express* **29**, 2940–2952 (2021)

Page 6:

Original:

...As a result, when V_g was 1 V, an extremely large responsivity of 2.1×10^6 A/W was obtained at an incident power of 631 fW.

Revised:

...As a result, when V_g was 1 V, an extremely large responsivity of **approximately 1×10^6 A/W** was obtained at an incident power **between 631 fW – 6.31 pW**.

Page 9:

The noise equivalent power (NEP) and specific detectivity were also evaluated from the measured noise power density spectrum when V_d was 0.5 V (see Supplementary Section V). As shown in Fig. 6, the measured NEP took a minimum value when V_g was 0 V owing to the balance between the dark current and responsivity. As a result, the specific detectivity exceeded 1×10^{12} cm Hz^{1/2} W⁻¹, which is approximately 100 times greater than that of a Ge photodetector⁵⁵. Hence, our phototransistor exhibits

high sensitivity as well as an ultrahigh responsivity.

Fig. 6

Section VI in Supplementary

The noise power density spectrum of an InGaAs phototransistor was evaluated by the Fourier transform of the dark current waveform measured using a waveform generator/fast measurement unit (Agilent, B1530A) of a semiconductor device analyzer (Agilent Technologies, B1500A). Figure S10 shows the measured noise power density spectrum when V_d and V_g were 0.5 V and 1.0 V, respectively. As shown in Fig. S10, the noise power density was proportional to $1/f^2$, where f is a frequency. According to the method described in Ref. 4, the noise equivalent power (NEP) was extracted by integrating the noise power density from 0.1 Hz to 10 kHz. The specific detectivity was then obtained from the NEP, where the area of the phototransistor was the product of the waveguide width (0.4 μm) and InGaAs length (30 μm).

Fig. S10. Measured noise power density of waveguide-coupled InGaAs phototransistor. V_d and V_g was 0.5 V and 1.0 V, respectively.

Reference:

4. Weng, W. Y., Chang, S. J., Hsu, C. L. & Hsueh, T. J. A ZnO-nanowire phototransistor prepared on glass substrates. *ACS Appl. Mater. Interfaces* **3**, 162–166 (2011).

Comment #5

2 / The inserted power is not correctly measured: Fig. S5 does not show an exponential decrease with the length of the waveguide.

Response:

Thank you for this comment. Since the vertical axis is plot with a logarithm scale, the measured insertion loss (dB) should be linear with respect to InGaAs length. We have updated the insertion loss measurement and moved the discussion from Supplementary to the main manuscript to clarify this point.

Page 9:

The insertion loss of the InGaAs photodetector with varied lengths was evaluated as shown in Fig. 7, taking into account the propagation loss of the Si waveguide and the coupling loss of the grating coupler. As expected, the insertion loss in the log scale was proportional to the InGaAs length. Since

the insertion loss per unit length is 0.20 dB/ μm , we expect that the insertion loss can be less than 0.1 dB with the 0.5- μm -long phototransistor. As the device length decreases, the dark current also decreases. In addition, the total number of holes for achieving the same threshold voltage shift also decreases since the volume of the channel decreases with the device length. Thus, the responsivity of the scaled phototransistor is expected not to degrade markedly. Because of the high responsivity of the phototransistor, we can use it as an optical power monitor even with the device length of much smaller than 0.5 μm . Therefore, the proposed waveguide-coupled InGaAs phototransistor can potentially be used as a transparent optical power monitor for a Si waveguide.

Fig. 7. Insertion loss of the phototransistor as a function of InGaAs length.

Comment #6

In the introduction (lines 38-40) the authors explain that they aim to integrate “a numerous optical power monitors” in the optical circuit in order to provide a reliable method for configuring the phase shifters. Considering both the electrical power consumption of the photoFET and its absorption (optical losses), what is the limiting density of such devices in an optical circuit?

Response:

Thank you for the insightful comment. As the response to Comment #5, the insertion loss can potentially be reduced to 0.1 dB. If the total optical loss of 10 dB is acceptable, up to 100 photoFETs can be cascaded, enabling a 100×100 Si programmable photonic circuit. When the photoFET is biased with $V_g = 1$ V and $V_d = 0.5$ V, the transistor current is approximately 40 μ A, meaning that the power consumption of the single photoFET is approximately 20 μ W. In the case of a 100×100 Si programmable photonic circuit where approximately 10000 photoFETs should be required, the total power consumption of the photoFETs is approximately 200 mW. This power consumption can be reduced further by reducing the device length. Therefore, the power consumption does not limit the device density. We have added this discussion in the revised manuscript.

Page 10:

Original:

...Owing to the high responsivity, the proposed phototransistor can be a transparent in-line optical power monitor for a Si waveguide realized by reducing the length of the InGaAs absorber to less than 1 μ m (see Supplementary Section V).

Revised:

...Owing to the high responsivity, the proposed phototransistor can be a transparent in-line optical power monitor for a Si waveguide realized by reducing the length of the InGaAs absorber to less than 0.5 μ m (~~see Supplementary Section V~~). ~~If 10 dB insertion loss is acceptable, up to 100 photoFETs can be cascaded, enabling a 100×100 Si programmable photonic circuit. When the photoFET is biased with $V_g = 1$ V and $V_d = 0.5$ V, the transistor current is approximately 40 μ A, meaning that the power consumption of the single photoFET is approximately 20 μ W. In the case of a 100×100 Si programmable photonic circuit where approximately 10000 photoFETs should be required, the total power consumption of the photoFETs is approximately 200 mW. This power consumption can be reduced further by reducing the device length. Therefore, the power consumption does not limit the device density.~~

Comment #7

Figures 2 show only two curves (two values of V_d in Fig. 2a, two values of V_g in Fig. 2b). This is not enough to provide the reader with an understanding of the device behavior in the dark.

In addition, the x-label on Fig 2b is wrong. It must be “Drain voltage” instead of “Gate voltage”.

Response:

Thank you for this comment. As the reviewer pointed out, the x-label on Fig. 2b should be “Drain voltage”. We have revised Fig. 2b. As a I_d - V_g plot of Fig. 2a, it is quite common to show two I_d curves with low V_d and high V_d . As a I_d - V_d plot of Fig. 2b, we show I_d curves with $V_g = -1$ V, 0 V and 1V, sufficient for understanding the following discussion with the V_g range discussed in this paper.

REVIEWERS' COMMENTS

Reviewer #1 (Remarks to the Author):

I have now thoroughly read the comments and remarks from the previous evaluators and mapped those to the replies from the authors. Additionally, I have also perused the manuscript with great interest.

Summarized, I think this is a very well written manuscript reporting on an ultra-high responsivity waveguide coupled phototransistor operating at 1.3 μm . The concept of coupling the evanescent field from the waveguide to the inverted InGaAs channel is really nice, and the detector also demonstrates excellent performance. The authors have responded very well to all concerns raised by the other two reviewers. One small comment is that it is mentioned by the other reviewers that the results, before revision, were unsuitable for publishing in Nature Photonics. Has this paper originally been sent to Nature Photonics, and later being transferred to Nature Communications?

In any case, the paper is in my opinion now suitable for Nature Communications.

Reviewer #2 (Remarks to the Author):

The authors have done an excellent job in addressing the comments from both reviewers.

The manuscript itself is well-written and the results would be of great interest to researchers working in the field of Si photonics.

We would like to thank all the reviewers for kindly reviewing our paper and the editors for pointing out the editorial issues. We have revised our manuscript carefully in response to the editorial requests. The revised parts are highlighted in red color. Please find the details in the authors' response to each specific comment.

Sincerely,

Mitsuru Takenaka, Professor
Department of Electrical Engineering and Information Systems,
The University of Tokyo,
7-3-1 Hongo, Bunkyo-ku, Tokyo 113-8656, Japan
E-mail : takenaka@mosfet.t.u-tokyo.ac.jp
Phone : +81-3-5841-6733
Fax : +81-3-5841-8564

Reviewers' Comments:

#####

Reviewer #1 (Remarks to the Author):

I have now thoroughly read the comments and remarks from the previous evaluators and mapped those to the replies from the authors. Additionally, I have also perused the manuscript with great interest. Summarized, I think this is a very well written manuscript reporting on an ultra-high responsivity waveguide coupled phototransistor operating at 1.3 um. The concept of coupling the evanescent field from the waveguide to the inverted InGaAs channel is really nice, and the detector also demonstrates excellent performance. The authors have responded very well to all concerns raised by the other two reviewers. One small comment is that it is mentioned by the other reviewers that the results, before revision, were unsuitable for publishing in Nature Photonics. Has this paper originally been sent to Nature Photonics, and later being transferred to Nature Communications?
In any case, the paper is in my opinion now suitable for Nature Communications.

Response:

Thank you for the positive comments. This paper was originally submitted to Nature photonics and transferred to Nature Communications. We are very happy to hear that our revised paper is finally suitable for Nature Communications.

Reviewer #2 (Remarks to the Author):

The authors have done an excellent job in addressing the comments from both reviewers. The manuscript itself is well-written and the results would be of great interest to researchers working in the field of Si photonics.

Response:

Thank you for kindly reviewing our manuscript. We are very happy to share our results through the publication in Nature Communications.

Editorial requests:

Author information:

Request #1

Please review your complete author list to verify that it is complete and accurate. We ask that you consult with your coauthors to ensure that all names, affiliations, and titles are represented correctly. Note that if any authors are added or removed after this point then all authors will be requested to provide approval documentation that could potentially delay the production of your paper.

Response:

We confirmed all the author names and affiliations.

Article structure:

Request #2

We can accommodate up to 10 display items (Figures or Tables) in the main article. Each Figure and Table must fit easily within an A4 page (210 x 297 mm). Please ensure that the number and size of your Figures and Tables fulfil these requirements to avoid any delay in the acceptance of your article.

Response:

We confirmed the number of figures and adjust the size of all the figures.

To comply with this format and optimise the presentation of data in your Article, we suggest the following changes to the display items in your paper:

Request #3

Please make the scale bar more visible in figure 1b.

Response:

We made the scale bar more visible.

Request #4

Please ensure your main manuscript file includes the following sections, in this order:

Title

Author list

Affiliations

Abstract

Introduction

Results

Discussion (optional)

Results and Discussion (optional)

Methods

Data Availability

Code Availability (if relevant)

References

Acknowledgements

Author Contributions Statement

Competing Interests Statement

Tables

Figure Legends/Captions (for main text figures)

We do not edit Supplementary Information files; they will be uploaded with the published article as they are submitted with the final version of your manuscript. Any tracked changes should be removed from the file and the file should be provided as a PDF file. Supplementary Figures do not need to be provided separately.

Response:

We added “Results”, “Discussion”, and “Code availability” as well as “Introduction” as sections.

Main text:

Request #5

Please rearrange the Introduction so that all discussion of previous work appears first. The final paragraph should contain only a concise summary of the current work, in the present tense, and begin with a phrase like “In this work” or “Here, we show”.

Response:

We rearranged the manuscript so that all discussion of previous work appears first. In the last paragraph, we described the simple summary of our work.

Request #6

Please divide the Results section into subsections, each with a title of 60 characters or fewer including spaces.

Response:

We divided the results section into “Device structure and fabrication”, “Transistor characteristics”, “Photoresponses”, and “Time responses”.

Request #7

Please provide a Discussion section in the main manuscript file.

Response:

We added “Discussion” section in which we discussed the benchmark and the insertion loss of our device.

Request #8

Please do not use italics, bold font, underlining or speech marks except in headings unless required for technical terms (in both the main text and the display items).

Response:

We confirmed that we don’t use italics, bold font, underlining or speech marks except in headings.

Request #9

Please make sure that mathematical terms throughout your manuscript and Supplementary Information (including in figures, figure axes, and legends) conform strictly to the following guidelines. Equations must be supplied in editable format, and not as images. Scalar variables (e.g. x , V , χ) must be typeset in italic, whereas multi-letter variables and functions (e.g. \log) must be formatted in roman. Vectors (such as the wavevector k or the magnetic field vector B) must be typeset in bold without italics.

Response:

We confirmed this point in the manuscript and supplementary by revising the text and figures.

Request #10

Please label equations sequentially as (1), (2), (3), etc.

Response:

We added the equation number in Supplementary.

Figures and Tables:

Request #11

Please see the guidelines linked below for detailed instructions about how your figures should be prepared. Following these instructions will reduce the chances of delays should we need to request replacement artwork from you at a later stage.

<https://www.nature.com/documents/NRJs-guide-to-preparing-final-artwork.pdf>

Response:

We prepared all figures with editable formats using Powerpoint and followed the instructions.

Request #12

Figures must be accompanied by a legend of up to 350 words, referring to all panels within the figure.

Response:

We confirmed this point.

Request #13

Any abbreviations, symbols or colours present in your figures must be defined in the associated legends.

Response:

We confirmed this point.

Data and Code:

Request #15

Nature journals strongly support public availability of data and code. Please deposit the data and code used in your paper into a public data repository, or alternatively, present the data as Supplementary Information. If data can only be shared on request, please explain why in your Data Availability Statement, and also in the correspondence with your editor.

Please note that for some data types, deposition in a public repository is mandatory. Any restrictions on sharing of these data types must be clearly indicated in the statement and discussed with the editor. More information on our data deposition policies and available repositories can be found here: <https://www.nature.com/nature-research/editorial-policies/reporting-standards#availability-of-data>

Response:

We added the following statements as “Data availability” and “Code availability”.

The data that support the findings of this study are available from the corresponding authors on reasonable request.

The simulation and computational codes for this study are available from the corresponding authors on reasonable request.

Request #15

All published manuscripts reporting original research in Nature Portfolio journals must include a data availability statement, within the Methods and under the heading 'Data Availability'.

The data availability statement must make the conditions of access to the “minimum dataset” that are necessary to interpret, verify and extend the research in the article, transparent to readers.

This minimum dataset may be provided through deposition in public community/discipline-specific repositories, custom proprietary repositories or general repositories like Figshare, Zenodo and Dryad. Providing large datasets in supplementary information is strongly discouraged and the preferred approach is to make data available in repositories. Scientific Data, a Nature Portfolio journal, maintains a list of approved and recommended data repositories to support researchers seeking suitable repositories for their data (<https://www.nature.com/sdata/policies/repositories>).

The Data Availability Statement should also reference any source data published alongside the paper.

If DOIs are provided, we also strongly encourage including these in the Reference list (authors, title, publisher (repository name), identifier, year).

For clinical datasets or third party data, please ensure that the statement adheres to our policy (<https://www.nature.com/nature-research/editorial-policies/reporting-standards#availability-of-data>)

Response:

We added the following statements as “Data availability” and “Code availability”.

The data that support the findings of this study are available from the corresponding authors on reasonable request.

The simulation and computational codes for this study are available from the corresponding authors on reasonable request.

Request #16

Please use the following template to provide all the information stated above:

The XX data generated in this study have been deposited in the YY database under accession code ZZ [add hyperlink here]. The XX data are available under restricted access for {insert reason}, access can be obtained by {explain how}. The raw XX data are protected and are not available due to data privacy laws. The processed XX data are available at YY. The XX data generated in this study are provided in the Supplementary Information/Source Data file. The XX data used in this study are available in the YY database under accession code ZZ [Add hyperlink here].

Response:

We added the following statements as “Data availability” and “Code availability”.

The data that support the findings of this study are available from the corresponding authors on reasonable request.

The simulation and computational codes for this study are available from the corresponding authors on reasonable request.

Methods:

Request #17

Sufficient details of the experiments must be provided in the Methods section such that they could be reproduced without reference to published papers. Use of the term "as described previously" is not encouraged.

Response:

We provided the details of the experiments in the Methods section with Supplementary.

References:

Request #18

Supplementary References should appear at the end of the Supplementary Information file, and must be self-contained and numbered from 1. References mentioned in both the main text and the Supplementary Information should be part of both reference lists so that the Supplementary Information does not refer to the reference list in the main paper and vice versa.

Response:

We confirmed this point.

End matter:

Request #20

Nature Portfolio defines Competing Interest (CI) as financial and non-financial interests (including but not limited to funding, employment, stocks, shares, patents, personal or professional relationships with individuals or institutions, and unpaid membership advocacy) that could be perceived to directly

undermine the objectivity, integrity, and value of a publication, or could be seen as having an influence on the judgments and actions of authors with regard to objective data presentation, analysis, and interpretation.

Please thoroughly review our policy on Competing Interests and include a detailed statement both in your final manuscript file and in our manuscript tracking system. Please ensure the statements are identical in both. Be specific about how each point stated relates to the research and list applicable author initials, and/or patent numbers.

If there are no competing interests, a negative statement must be included.

<https://www.nature.com/nature-research/editorial-policies/competing-interests>

Response:

We confirmed this point.

Request #20

Please confirm that all relevant funding awarded to each author is described in the Acknowledgements section. List each grant number, followed by the initials of the author who received it.

Response:

We added the initial of the author receiving the grants.

Preparing your manuscript files:

Request #20

Unless otherwise stated please limit individual file sizes to approximately 30MB. We strongly encourage the use of repositories for large datasets or source data due to size considerations.

Response:

We confirmed that the size of each file is less than 30MB.

Request #21

Please supply a brief (maximum 250 characters, including spaces) summary of the main findings of the paper to be used on our website and in our e-alerts. The summary should be written in the third

person in language suitable for a broad audience. The summary may be edited by the editors prior to publication. Please provide this summary in your cover letter.

Response:

We provided the summary below in the cover letter.

The authors presented an ultrahigh-responsivity phototransistor with an ultrathin InGaAs on a Si waveguide. The effective gating by the Si waveguide enables 10^6 A/W responsivity, promising for optical power monitors in Si photonic circuits.

Request #22

To ensure maximum visibility for your work, we may tweet about your paper following publication. If you would like us to include the Twitter handles of the first author(s), corresponding author(s), lab or institution in this tweet, please provide them in your cover letter. We would also welcome your suggestions for hashtags to use when tweeting about the work.

Response:

We provided the Twitter account of the corresponding author.

@takenaka326

Request #23

Please supply the main manuscript file in either Microsoft Word or LaTeX format

Response:

We prepared the main manuscript using Microsoft Word.

Request #24

Please provide figures as individual vector files with editable text. Acceptable file types for figures are .ai, .eps, .pdf, .ppt or Chem Draw for fully editable vector-based art. For detailed guidance on figure preparation, see <https://www.nature.com/documents/aj-artworkguidelines.pdf>

Response:

We provided figures using Microsoft Powerpoint.

Request #25

The use or adaptation of previously published images is strongly discouraged. If this is unavoidable, please request the necessary rights documentation to re-use such material from the relevant copyright holders and return this to us when you submit your revised manuscript. Please check whether your manuscript or Supplementary Information contain third-party images, such as figures from the literature, stock photos, clip art or commercial satellite and map data.

For more information on what constitutes ownership by a third party, please contact our Editorial Assistant at naturecommunications@nature.com

Response:

We have no images from the previous publications.

Forms to complete:

Request #26

Editorial Policy Checklist

Please update and upload a final version of the Editorial Policy Checklist with your revised manuscript files. A blank Editorial Policy Checklist can be found via the link below. Note that this form is a dynamic ‘smart pdf’ and must be downloaded and completed in Adobe Reader.

Please update your current checklist or download from:

<https://www.nature.com/documents/nr-editorial-policy-checklist.zip>

Response:

We updated the checklist and will upload it.

You will need to upload:

Request #27

Editorial Policy Checklist

Completed Third Party Rights Table (if relevant)

A completed copy of this checklist

The main manuscript file in either Microsoft Word or LaTeX format

Separate Figure files

Inventory of Supporting Information
A Supplementary Information file

Response:

We will upload these files.